# Psychological Care in Spanish Nurses at the Frontline of the COVID-19 Pandemic: A Prospective Study on Symptoms, Burnout and Psychological Variables

**DOI:** 10.3390/healthcare13101108

**Published:** 2025-05-09

**Authors:** Cecilia Peñacoba-Puente, Fernando José García-Hedrera, Mercedes Gómez-Del-Pulgar García-Madrid, Francisco Javier Carmona-Monge, Fernanda Gil-Almagro

**Affiliations:** 1Health Sciences Faculty, Department of Psychology, Psychology, Rey Juan Carlos University, Av. de Atenas, s/n, 28922 Alcorcón, Madrid, Spain; cecilia.penacoba@urjc.es; 2Nurse Intensive Care Unit, Hospital Universitario Fundación Alcorcón, Budapest, 1, 28922 Alcorcón, Madrid, Spain; fjgarciah@gmail.com; 3Department of Simulation, Francisco de Vitoria University in Pozuelo, M-515, km 1, 800, 28223 Pozuelo de Alarcón, Madrid, Spain; m.gomezdelpulgar@ufv.es; 4Anesthesia Department, Hospital Universitario Santiago de Compostela, Rúa da Choupana, s/n, 15706 Santiago de Compostela, A Coruña, Spain

**Keywords:** anxiety, nurses, pandemic, psychological help, cognitive fusion, burnout

## Abstract

(1) Background: Previous studies have highlighted the emotional symptoms experienced throughout the COVID-19 pandemic by nurses and their consequences. It would be of interest to analyze the extent to which healthcare workers (HCWs), in the context of the psychological health crisis, have sought (and received) psychological care. Likewise, it would be highly relevant to analyze the profile of these professionals, both in terms of the sociodemographic and occupational characteristics as well as the emotional symptoms they presented, and the percentage of nurses who requested psychological help during the COVID-19 pandemic, their sociodemographic and occupational characteristics as well as their levels of associated symptoms. Additionally, one could study the associated psychological personality variables, including both risk factors and protective variables, as this is of special interest for the design of appropriate interventions. (2) Methods: An observational, descriptive, prospective longitudinal study with three data collection periods was carried out. At the first time point, anxiety, insomnia, self-efficacy, resilience and social support were assessed. Anxiety, insomnia, fear of COVID-19, cognitive fusion and burnout syndrome were assessed at the second time point. Finally, at the third time point, we assessed anxiety, insomnia and burnout syndrome. During the second and third time points, the nurses’ requests for psychological help were assessed. (3) Results: Overall, 33.1% of the final sample requested psychological support, and 20.5% of them had sought psychological support by the first time point, of which 7.3% continued to in the final time measure. The request for psychological help was significantly related to higher means for anxiety (*p* = 0.003), insomnia (*p* = 0.001) and burnout (*p* < 0.05), as well as high levels of cognitive fusion (*p* = 0.001) and low levels of resilience (*p* = 0.009). Requests for psychological help were not significantly related to social support (*p* = 0.222) or fear of COVID-19 (*p* = 0.625). (4) Conclusions: The data suggest the need to promote measures for the implementation of psychological help among nurses aimed not only at reducing the consequences of the psycho-emotional affectation derived from a stressful work situation but also strengthening health-promoting traits such as self-efficacy or resilience.

## 1. Introduction

There is extensive research highlighting the significant psycho-emotional distress experienced by nurses during the aftermath of the COVID-19 pandemic [1,2]. These studies delved into different risk factors, including occupational aspects such as work overload, working at the bedside of infectious patients or fear of contagion [3]. Other studies have explored the sociodemographic or psychosocial factors of the professionals themselves, such as age, gender, resilience or self-efficacy [4,5]. Previous research has also shown the clinical implication of exposure to these highly stressful situations, leading to anxiety disorders, depression or burnout [6,7].

There is no doubt that almost four years after the beginning of the pandemic, it is of special interest to analyze the evolution of these symptoms in the medium and long term, a period that has come to be known as the “post-pandemic stage”. Different studies, using prospective designs, have observed decreasing trends in anxiety and stress but have pointed to increases in burnout and specifically an increase in the depersonalization of HCWs [8]. Burnout syndrome has been of particular concern to researchers in the context of the COVID-19 pandemic because it is frequent among HCWs [9] and because it often appears in the presence of certain associated risk factors that have increased during this crisis, such as shift work, increased burden of care and fear of making mistakes in healthcare practice [10]. In addition, during the COVID-19 pandemic, burnout has been studied in nurses as a group at special risk, showing an increase in levels of emotional exhaustion and depersonalization, as well as a clear decrease in personal accomplishment [11,12]. The above research has also provided interesting data on individual variability in psycho-emotional distress following exposure to a highly stressful situation, such as the COVID-19 pandemic [13].

These results, derived from the conceptualization of interactionist models of stress [14], highlight the vision of individuals as agents of their health and fortunately move away from traditional deterministic models, where the individual was understood as a mere victim of events. Along these lines, in the context of the impact of the COVID-19 pandemic among HCWs, research has shown the relevance of certain personality traits and processes as additional risk factors for disease or as protective factors for health, highlighting the role of resilience [4].

The previous literature has shown that certain resilience factors can generate positive consequences and the development of strengths after experiencing stressful work situations [15]. Specifically, when analyzing the positive consequences of fostering resilience in HCWs, decreases in anxiety and burnout, especially in emotional exhaustion, were observed [16]. Furthermore, it was also found that personalities were shown to be more resilient after a traumatic work event. Another protective personality trait analyzed in nurses during the COVID-19 pandemic was perceived self-efficacy. Research conducted during the pandemic shows that high levels of self-efficacy are associated with positive consequences, such as decreased psycho-emotional symptoms including stress, anxiety or insomnia and the development of psychological strengths (such as a resilient personality) after a traumatic work experience [17,18].

Regarding the risk factors for the disease, the most frequently mentioned ones are fear (specifically experienced in the face of COVID-19) and cognitive fusion [3]. These factors have been suggested to increase the chronification or negative psycho-emotional consequences derived from stressful work situations. Fear of COVID-19 has been associated with high levels of stress and anxiety, high levels of fatigue, younger ages and the female gender [19]. Regarding cognitive fusion, understood as the difficulty of detaching oneself from one’s own thoughts, it has been shown to have a role as a chronifying factor of negative consequences after a stressful work situation [20]; positive correlations have been observed with anxious and depressive symptoms [21].

In this context, taking into account the abundant research that shows the unquestionable impact of the COVID-19 pandemic on the mental health of HCWs, it would be of interest to analyze the extent to which these professionals have requested (and received) psychological care. It would also be of interest to analyze their profile, both in terms of sociodemographic and occupational characteristics and the emotional symptoms they presented. In contrast to the abundance of research analyzing the mental health consequences of the COVID-19 pandemic on HCWs, and nursing personnel in particular, there have been rather few studies analyzing how many of these professionals have requested and received psychological help and the reasons that led them to do so. One such study [22] conducted among HCWs in Mexico who sought psychological help due to the COVID-19 pandemic showed that nearly half of them were at risk of suicide, with nurses being affected the most, showing that secondary traumatic stress, high negative affectivity, low positive affectivity, emotional insecurity and interpersonal problems were some of the risk factors [22].

Another study examined the use of formal and informal psychological support resources at the workplace among HCWs during the first and third waves of the COVID-19 pandemic in Ireland [23], highlighting the importance of psychological resources in the management of highly stressful situations and emphasizing peer support as the main resource. To the best of our knowledge, no studies have been carried out among Spanish nurses that delved into this issue. This type of study is especially relevant since previous research has shown the importance psycho-emotional interventions to reduce the consequences of a traumatic event [24]. Specifically, evidence shows that interventions, such as mindfulness, are positive influences in reducing anxiety, stress and burnout in nurses [25]. Therefore, it is of importance to analyze, in this context, the request for psychological help by nurses, as well as their profiles and associated psychological variables and symptoms, especially if we take into account that, despite the undoubted need for psychological help during the COVID-19 pandemic, most of them consider that they have no need for it [26].

In this context, the present study aims to analyze the request for psychological help by Spanish nurses during the COVID-19 pandemic, using a prospective longitudinal design with three periods of data collection from the beginning of the pandemic (period of confinement in Spain) until 18 months later. The present study aims to analyze the percentage of nurses who requested psychological help during the pandemic, their sociodemographic and occupational characteristics and their levels of associated symptoms. In addition, and of special interest for the design of appropriate interventions, there has been a focus on the study of associated personality variables, including both risk and protective factors.

## 2. Materials and Methods

### 2.1. Design

This study was based on a quantitative research approach, using a prospective longitudinal observational design to analyze changes and associations across three time points. No experimental manipulation was applied, and data were collected through validated psychometric self-report instruments. The research followed the STROBE guidelines for observational studies to ensure methodological rigor and transparency. Within this prospective nature, three phases of data collection in the context of the COVID-19 pandemic were analyzed. These phases took place as follows: (1) from 1 June to 21 June 2020, corresponding to the final stage of the state of alarm declared in Spain on 14 March; (2) a follow-up conducted six months after the end of the state of alarm (between January and March 2021); and (3) a third evaluation one year later (from April to July 2022). In the first phase, data were gathered on the sociodemographic and occupational characteristics, along with measures of anxiety, insomnia, self-efficacy, resilience and perceived social support. The second phase focused on evaluating anxiety, insomnia, fear related to COVID-19, cognitive fusion and burnout syndrome (including its three dimensions: emotional exhaustion, depersonalization and reduced personal accomplishment). During the third phase, assessments of anxiety, insomnia and the three dimensions of burnout syndrome were repeated. Additionally, in both the second and third phases, the study recorded whether nurses had sought psychological support.

#### 2.1.1. Procedure

The data were gathered through an electronic questionnaire created by the research team using the Google Forms platform which was specifically designed for this study. At the beginning of the form, participants were informed about the study’s objectives and asked to provide their informed consent for the use of their data. The survey link was distributed to nurses working in the Spanish healthcare system who had been in contact with COVID-19 patients. It was shared via social media platforms (such as Facebook, Twitter, LinkedIn and WhatsApp), and also through institutional emails from both public and private healthcare centers in Spain. The first round of data collection took place from 1 June to 21 June 2020. For the second round, carried out between January and March 2021, researchers contacted the same group of nurses who had participated in the initial phase, using their email addresses and inviting them to take part again. In the third and final data collection period, conducted between April and July 2022, emails were sent once more to those who had taken part in the earlier phases, requesting their continued collaboration in the study.

#### 2.1.2. Participants

The sample was composed of nurses working in the Spanish healthcare system. The sample was selected using non-probabilistic convenience sampling. A minimum sample size of n = 120 was considered for prospective mediation studies [27]. The following inclusion criteria were taken into account: working as a nurse during the data collection period and being in direct contact with COVID-19 patients. The following were considered as exclusion criteria: change of service during the study period or working as a nurse manager.

With awareness of the usual sample loss in longitudinal studies with this population (HCWs), in addition to the contextual conditions of data collection (COVID-19 pandemic) [28,29], at the first time point, a minimum sample size of 300 participants was established, obtaining a final sample of 534. Of these, at the second time point 6 months later, the participation of 331 nurses was obtained, and for the third time point one year later, a sample of 151 nurses was obtained, with this being the final sample of the study.

It should be noted that no statistically significant differences were observed in the variables of interest between the nurses who did not complete the three phases of the study (n = 383) and those who completed the study (n = 151).

### 2.2. Variables and Instruments

#### 2.2.1. Sociodemographic and Occupational Variables

Sociodemographic variables (gender, age and marital status) and occupational variables (employment status, years of experience in the service and the service in which they performed their duties) were collected using an ad hoc instrument developed by the research team.

To analyze requests for psychological help, an ad hoc question was posed with a yes-or-no response format: “Have you requested psychological help at any time during these months due to the effects of the pandemic?” This variable was analyzed during the second and third time points.

#### 2.2.2. Variables in Relation to Symptoms, Burnout Syndrome and Personality

(a)Symptoms-Generalized anxiety [assessed at time points 1, 2 and 3]: Symptoms of generalized anxiety disorder were evaluated using the Spanish version of the Generalized Anxiety Disorder Scale (GAD-7) [30,31]. This instrument includes 7 items with a 4-point Likert response scale, ranging from 0 (not at all) to 3 (nearly every day). Higher total scores reflect increased severity of anxiety symptoms. In the current study, the scale demonstrated excellent internal consistency, with a Cronbach’s alpha of 0.93.-Insomnia [assessed at time points 1, 2 and 3]: To evaluate insomnia-related symptoms, this study employed the Spanish version of the Insomnia Severity Index (ISI) [32,33]. This brief questionnaire is aligned with the diagnostic criteria outlined in the Diagnostic and Statistical Manual of Mental Disorders and the International Classification of Sleep Disorders. It consists of 7 items that assess three dimensions: severity, impact and satisfaction. Responses are given on a Likert-type scale ranging from 0 (no difficulty) to 4 (extremely severe difficulty), resulting in a total score between 0 and 28. A score of 22 or above indicates severe clinical insomnia. In the current study, the scale demonstrated good internal consistency, with a Cronbach’s alpha of 0.873.(b)Burnout Syndrome-Burnout [assessed at time points 2 and 3]: The Spanish adaptation of the Maslach Burnout Inventory–Human Services Survey (MBI-HSS) was utilized in this study [34,35]. This instrument consists of 22 items rated on a 7-point Likert scale, ranging from 0 (never) to 6 (every day). It evaluates three core dimensions of burnout: emotional exhaustion, depersonalization, and reduced personal accomplishment. In the present research, the overall internal consistency of the scale was high, with a Cronbach’s alpha of 0.88. The reliability coefficients for the individual subscales were 0.90 for emotional exhaustion, 0.72 for depersonalization, and 0.84 for reduced personal accomplishment.(c)Personality-Resilience [assessed at time point 1]: This study used the Spanish version of the Resilience Scale (RS-14) [36], which includes 14 items rated on a 7-point Likert scale, ranging from 1 (strongly disagree) to 7 (strongly agree). The total score spans from 14 to 98, where higher values reflect a greater level of resilience. In this study, the instrument showed excellent internal consistency, with a Cronbach’s alpha of 0.94.-Self-efficacy [assessed at time point 1]: The Spanish version of the General Self-Efficacy Scale (GSES) [37,38] was employed to assess individuals’ perceived ability to cope with various life challenges. The scale consists of 10 items, each rated on a 4-point Likert scale ranging from 1 (“not at all true”) to 4 (“completely true”). The total scores fall between 10 and 40, with higher scores reflecting greater self-efficacy. In the present study, the scale demonstrated good internal consistency, with a Cronbach’s alpha of 0.86.-Cognitive fusion [assessed at time point 2]: The Spanish version of the Cognitive Fusion Questionnaire (CFQ) [39,40] was utilized in this study. This instrument includes 7 items designed to measure the degree of cognitive fusion, or how strongly individuals are psychologically influenced or dominated by the content and form of their own thoughts. Responses are given on a 7-point Likert scale, ranging from 1 (never) to 7 (always). In our study, the questionnaire exhibited excellent internal consistency, with a Cronbach’s alpha of 0.97.-Fear of COVID [assessed at time point 2]: To measure this variable, this study employed the instrument developed by D. Ahorsu during the COVID-19 pandemic [41], in its Spanish version [42]. The scale consists of seven items and demonstrates a stable unidimensional structure along with strong psychometric properties. It uses a 5-point Likert response format, ranging from 1 (strongly disagree) to 5 (strongly agree) and yielding a total score between 7 and 35. Higher scores reflect a greater level of fear related to COVID-19. In our sample, the instrument demonstrated good internal consistency, with a Cronbach’s alpha of 0.87.-Social support [assessed at time points 2 and 3]: This study utilized the Spanish version [43] of the Multidimensional Scale of Perceived Social Support (MSPSS) [44]. This tool consists of 12 items designed to assess perceived social support across three sources—family, friends, and significant others—with each subscale comprising 4 items. An overall social support score is calculated as the average of these three dimensions. Items are rated on a 7-point Likert scale ranging from 1 (strongly disagree) to 7 (strongly agree). The MSPSS has demonstrated solid reliability across multiple studies. In this research, the total perceived social support scale showed excellent internal consistency, with a Cronbach’s alpha of 0.95.

**Data analysis:** Data analysis was performed using IBM SPSS Statistics version 27.0 (IBM Corp., Armonk, NY, USA). Descriptive statistics and reliability analysis using Cronbach’s alpha were conducted for all variables included in the study. Categorical variables were summarized using frequencies (n) and percentages (%), while continuous variables were described using means and standard deviations (SDs). To examine the bivariate relationships between variables, Pearson’s correlation was applied. Depending on the nature of the variables, chi-square tests (χ^2^), Student’s *t*-tests and one-way ANOVA were used to assess the associations. Multivariate analyses such as binary logistic regression were also used. A *p* value less than 0.05 was considered to indicate statistical significance.

## 3. Results

### 3.1. Characteristics of the Sample

A total of 151 nurses participated in the study at the three time points. As can be seen in Table 1, almost the entire sample was composed of women. The mean age of the participants was 41.36, with a minimum age of 21 and a maximum age of 62.

When analyzing the service, the majority of the sample was concentrated in intensive care units (ICUs) and hospitalization (41.1% and 32.5%, respectively), followed by emergency (17.2%) and primary care (PC) or consultations (9.3%). The mean number of years of experience in the service was 10.71 years, ranging from those who had just joined the profession up to a maximum of 35 years of experience.

### 3.2. Descriptive Analysis of Symptoms, Burnout Syndrome and Personality Variables

Table 2 shows the mean scores for the sample for the different variables. Beginning with symptoms, as far as anxiety is concerned, at the first time point, the nurses participating in the study showed mean scores compatible with moderate levels of anxiety. A clear improvement in anxiety was found over time, showing mean scores compatible with mild levels of anxiety at the third time point.

As for insomnia, at the first time point, the nurses showed mean scores compatible with symptoms of subclinical insomnia, with scores decreasing slightly during the second and third time points, although the scores continued to be compatible with subclinical insomnia.

Regarding burnout syndrome, evaluated at the second and third time points, different scores and evolutions were observed, depending on the subscale considered. Regarding emotional exhaustion, at the second time point, the nurses showed mean scores compatible with high emotional exhaustion, which decreased at the third time point, in which the scores indicated medium emotional exhaustion. As for depersonalization, at the second time point, the nurses showed medium means, but this trait evolved negatively, presenting higher means compatible with high depersonalization at the third time point. Regarding personal accomplishment, at the second time point, the nurses presented mean scores compatible with medium personal accomplishment. Similarly, this evolution was negative, as the results at the third time point showed averages compatible with low personal accomplishment.

The levels of self-efficacy and perceived social support in the sample at the first time point were compatible with medium levels, while resilience showed high levels. The levels of fear of COVID-19 were considered moderate. As for the cognitive fusion analyzed at the second time point, the nurses showed means compatible with levels of non-clinical cognitive fusion.

### 3.3. Request for Psychological Support

Regarding the request for psychological support by Spanish nurses, of the final sample (n = 151), a total of 50 (33.1%) requested psychological support. Specifically, 20.5% (n = 31) of them had sought psychological support by the first time point, of which 7.3% (n = 11) continued to do so in the final time measure, while 12.6% (n = 19) had requested it by the second time point.

### 3.4. Association Between the Request for Psychological Support and Sociodemographic and Occupational Variables

When analyzing the association between sociodemographic and occupational variables and the request for psychological support (Table 3), no statistically significant differences were found for the variables considered. Specifically, age did not appear to be related to requesting help (*p* = 0.185), with similar mean ages between those who requested support (mean = 39.35, SD = 9.06) and those who did not (mean = 41.88, SD = 9.50). Likewise, years of professional experience did not show a significant association (*p* = 0.450), with means of 9.61 years for those who sought help and 10.99 years for those who did not. Gender differences were also non-significant (*p* = 0.950), although women (86.8% of the sample) represented the majority of those who requested psychological support (21%), similar to their proportion in the total sample.

Regarding family status, there were no statistically significant differences in the request for support among married or cohabiting (19.8%), single (21.6%) or separated nurses (25%) (*p* = 0.924). When analyzing the healthcare service, although the results were not statistically significant (*p* = 0.442), it is noteworthy that nurses from primary care and consultations were the ones with the highest proportion of support requests (28.6%), followed by those in emergency units (26.9%) and hospitalization services (22.4%). ICU nurses, despite being the most represented group (41.1% of the sample), reported the lowest rate of psychological help requests (14.5%).

Finally, the only variable approaching statistical significance was employment status (*p* = 0.058), with interim nurses showing the highest proportion of support requests (30.2%), followed by those with temporary contracts (25.8%) and permanent staff showing the lowest rate (13.0%). This trend, although not statistically significant, may suggest a relationship between job stability and the likelihood of seeking psychological support, possibly due to increased perceived vulnerability among non-permanent staff.

### 3.5. Associations Between the Request for Psychological Support and Symptoms, Burnout Syndrome and Personality Variables

Table 4 shows the associations between the request for psychological support and the different variables considered (symptoms, burnout and personality). When analyzing the symptoms compatible with generalized anxiety, it was found that at the first time point, nurses who had requested psychological help during the pandemic had higher mean scores (mean = 13.30; SD = 4.69) compared with those who did not (mean = 10.28; SD = 6.19) (*p* = 0.003). These significant differences continued throughout the following time points, with the third time point showing that nurses who requested psychological help continued to have mean scores compatible with moderate anxiety compared with those who did not request psychological help, who had mean scores corresponding to mild anxiety (*p* = 0.000), indicating a sustained psychological impact in this subgroup.

When analyzing insomnia, it was observed that the nurses with mean scores compatible with moderate clinical insomnia at the first time point were the ones who requested psychological help compared with those who did not, who had mean scores compatible with subclinical insomnia (*p* = 0.001). The difference remined significant in the second (*p* = 0.010) and third measurements (*p* < 0.001), suggesting a chronic pattern of sleep disturbance among those requesting support.

In terms of burnout, emotional exhaustion was the dimension most strongly associated with seeking psychological help. At the second time point, nurses who sought help scored significantly higher for emotional exhaustion (mean = 33.42, SD = 12.40) than those who did not (mean = 25.60, SD = 13.18; *p* = 0.001), and this difference remained significant at the third time point (*p* = 0.006). Depersonalization also showed a significant difference (*p* = 0.040) regarding requesting psychological help at the second time point, though not at the third time point (*p* = 0.068). Personal accomplishment was lower in those nurses who requested help at the second time point (*p* = 0.040), although this difference was no longer significant at the third time point (*p* = 0.186). Social support and fear of COVID-19 did not present significant differences (*p* = 0.222 and *p* = 0.625, respectively) regarding requests for support, as we found extremely similar means between nurses who requested psychological help and those who did not. In relation to cognitive fusion, significant differences were found, as nurses who scored higher for cognitive fusion were the ones requesting psychological support (*p* = 0.001).

In addition, in order to analyze, among the possible statistically significant variables at the bivariate level, the predictor variables within the overall impact, a logistic regression analysis was carried out (enter method). The results show that anxiety experienced at the third time point was the only predictor variable (B = 0.225, SD = 0.047; Wald = 23.153, Exp (B) = 1.252, *p* < 0.001).

## 4. Discussion

The current study analyzed, by means of a prospective longitudinal study during three periods of data collection, requests for psychological help during the COVID-19 pandemic by nurses and their sociodemographic and occupational profiles, as well as the possible associations with the symptoms experienced, burnout syndrome and different psychosocial variables.

Regarding the sociodemographic profile, it is interesting to note that throughout this health crisis, numerous studies have shown age and gender as risk factors for psycho-emotional disorders after a stressful work situation, such as the COVID-19 pandemic, with younger staff and women being the most susceptible sociodemographic profile [45,46,47]. However, our study in Spanish nurses suggests that there are no significant differences in age or gender when it comes to requesting psychological help after a stressful work situation. These results, in the absence of new studies that delve deeper into this issue, provide interesting questions to reflect upon. Thus, while the previous literature established a mental health risk profile in HCWs linked to age (i.e., young people) and gender (i.e., women), our data indicate that this more vulnerable group is not characterized by a greater demand for psychological help.

Similarly, scientific evidence has shown that ICU personnel have been the ones hardest hit by the pandemic, presenting high levels of burnout and psychoemotional symptoms later on [1,48]. However, in our study, there were no significant differences in terms of the requests for psychological help and the service to which they belonged. Specifically, although the differences were not statistically significant, it was the ICU nurses who requested psychological help less frequently (about 15%) than nurses in other services (about 25%). Our data show that primary care (PC) nurses were the ones who requested psychological help the most. In this sense, it is necessary to bear in mind that there is little research regarding burnout suffered by PC nurses, making it of interest to focus on this group, since during the pandemic, studies on ICU professionals proliferated, but studies on PC personnel have been scarcer, although they have also shown prominent signs of psychoemotional distress [49].

Regarding the evolution of symptoms, the results of our longitudinal study, in line with the existing literature, show a clear improvement in symptoms such as anxiety and insomnia [28,50], presenting a decrease throughout the study. However, the symptoms derived from the burnout study only show improvement for the emotional exhaustion subscale, and the nurses at the third time point were more depersonalized and showed less personal accomplishment. According to the existing literature, this fact has already been recorded [11,51]. This result could be explained by the lack of protective or mediating measures implemented since the beginning of the pandemic, since although the existing studies included therapeutic interventions such as mindfulness, few healthcare centers have implemented measures to analyze and improve possible alterations suffered by HCWs [52,53].

When we analyzed the relationship between symptoms, burnout and personality variables and requesting psychological help, our findings highlighted the importance of establishing preventive or buffering measures for psycho-emotional damage after a stressful work situation, since it was the nurses suffering higher levels of affectation who requested psychological help.

In terms of anxiety and insomnia, nurses who sought psychological help had higher scores than the mean of the sample, indicating a clear psychoemotional affectation derived from symptoms compatible with moderate anxiety and subclinical insomnia. There is abundant research showing that nurses after the pandemic have presented elevated levels of anxiety and insomnia [54], suggesting, therefore, that coping strategies remain insufficient and poorly defined [55]. Our results also support the role of anxiety in the post-COVID-19 pandemic as the main predictor of seeking psychological support. Burnout has also been extensively studied in HCWs during the pandemic [56], but approaches to how to reduce or buffer burnout are still unclear. Some studies have highlighted the importance of the physical and mental well-being of nurses and proposed individual and organizational approaches to support the transition of novice nurses through accompaniment of an experienced nurse as a measure to reduce burnout [57].

It is interesting to analyze how the intervention of psychological help is beneficial in all spheres after the passage of a stressful situation [24]. However, some studies have shown that HCWs do not seek psychological help [58]. Other research showed that although HCWs were aware of their psychoemotional distress after the COVID-19 pandemic, most feel that they did not need psychological help [26]. Our study also showed a high level of psychoemotional disturbance in nurses, with long-term consequences such as increased depersonalization and low personal accomplishment, as well as emotional exhaustion. However, only one third of the sample (33.1%) requested psychological assistance.

In general, it is important to highlight that practically all research in this area insists on the need for health professionals to receive psychological support in the wake of the COVID-19 pandemic, yet there are hardly any studies analyzing how many professionals sought psychological help and what the characteristics of these professionals were [59,60,61]. In the few existing studies, none of them were carried out with a Spanish sample, but their results were similar to ours. Thus, for health professionals from Mexico and Paris who request psychological help, elevated symptoms, including indicators of anxiety, depression and stress, were confirmed [22,62]. Particularly relevant is the study carried out in Mexico [22], which showed that almost 50% of health professionals requesting psychological support were at risk of suicide and that nurses were the most vulnerable group.

The recent literature is beginning to reflect the quite worrying consequences of the pandemic on HCWs, such as the intention to leave the profession or even the risk of suicide derived from post-traumatic stress disorder and high levels of depression [22,63]. These consequences may be associated with high cognitive fusion or lack of resilience [64,65]. In this sense, our data indicate high levels of cognitive fusion in nurses requesting support and highlight that nurses requesting psycho-emotional help showed lower levels of resilience.

Social support, however, did not seem to be associated with requests for psychological help in our study. This may be due to the fact that during the COVID-19 pandemic, given the infectious nature of the pathology, social relationships were diminished by the fear of contagion [3,66].

Throughout the data collection period, a progressive decrease in requests for psychological support among nurses in Spain was observed. This trend may be explained by various factors. On the one hand, the stigma associated with seeking psychological help, which is particularly prevalent in Latin sociocultural contexts, may have influenced the low demand for these services. On the other hand, as the pandemic progressed, lockdown restrictions in Spain eased, and fear of COVID-19 subsided, which enabled greater access to social and familial support networks that were not available during the initial phases of the health crisis due to restrictions or fear of contagion. On the other hand, the effects of psychological therapy must be considered. Psychological support acts as a protective factor against emotional exhaustion and other mental health issues among healthcare professionals [23]. Therefore, it is essential to emphasize the importance of psychological support among healthcare workers as a key tool for maintaining psycho-emotional well-being.

Finally, it is necessary to point out some of the limitations of our research. Since this was a probability sample of convenience, the sample was mainly concentrated in Madrid and Catalonia, and the data from the rest of the autonomous communities of Spain are less representative. This limitation should be taken into account when generalizing the results. However, it should be noted that no statistically significant differences were found in the variables of interest between the different autonomous communities. On the other hand, an important limitation is the significant sample loss between the first, second and third time points. This sample loss has previously been found in this type of study, both with this group and during the COVID-19 pandemic [28,29]. The use of self-reporting questionnaires for the measurement of anxious symptoms can be considered a bias, although it is the most common and established form of assessment with adequate indicators of validity and reliability based on the assessment instrument used. The questionnaire to be completed included several instruments, which meant that it took a long time to complete (approximately 25 min). This could have been a significant conditioning factor, leading to an increase in sample size loss. Another limitation of this study is that burnout was not assessed at the initial time point of the pandemic. Including this measure from the beginning would have allowed for a more rigorous longitudinal analysis of its evolution over time. The absence of baseline data for burnout limited our ability to fully understand how this syndrome developed in response to prolonged exposure to pandemic-related stressors and restricted comparisons with other variables that were measured across all three time points.

Additionally, this study did not include organizational or institutional variables that are known to influence both psychological well-being and help-seeking behavior among healthcare professionals. Relevant factors such as workload, access to mental health resources, availability of institutional psychological support services or perceived leadership support were not assessed. The absence of these contextual variables limits the scope of the analysis, as these elements are well documented in the literature as critical contributors to burnout and the decision to seek psychological help. Future research should incorporate these dimensions in order to provide a more comprehensive understanding of the factors influencing mental health outcomes in healthcare settings.

Finally, it would have been of interest to take into account the emotional symptoms of the participants prior to the pandemic in order to include this variable.

## 5. Conclusions

We consider the present study, despite the above limitations, to have important practical implications. It is important to begin to establish psychological support measures for nursing personnel, characterized in the existing literature as the HCWs experiencing the greatest psycho-emotional impact during the pandemic [67]. These measures should be aimed at implementing psycho-emotional care for our professionals, in addition to advising them of the benefits of psycho-emotional therapies based on psychological support (cognitive-behavioral therapy or acceptance and commitment therapy) focused through an assessment of the needs of nurses [68,69] or group sessions supervised by psychologists [70]. There have been few studies examining the benefits of early psychological support, since most of them highlighted its need but did not establish the implementation of measures. However, those that described psychological support as a buffer therapy for negative consequences for HCWs pointed out its clear benefit [71].

The present study, through a prospective longitudinal observational design, showed that despite the fact that the psychoemotional symptoms of nurses derived from the COVID-19 pandemic generated important negative consequences, few nurses requested psychological help. In addition, seeking psychological help is associated with elevated levels of anxiety, insomnia, cognitive fusion and burnout, as well as low levels of resilience and self-efficacy. No associations were found between the request for psychological support and sociodemographic and occupational variables. The previous literature on the study of this issue is scarce, with only a few studies carried out on HCWs in Mexico [22], Ireland [23] and Paris [62]. To the best of our knowledge, our study is the first to be carried out with health professionals in Spain, specifically nurses. Thus, despite its novelty, future lines of research should delve deeper into the issues raised here.

In short, the data point to the need to promote measures for the implementation of psychological help in nurses aimed not only at reducing the consequences of the psycho-emotional affectation derived from a stressful work situation but also strengthening health-promoting traits such as self-efficacy or resilience.

## Figures and Tables

**Table 1 healthcare-13-01108-t001:** Descriptive of sociodemographic and professional variables (n = 151).

		n	%	Mean	SD
Age (years)				41.36	9.44
Service Experience				10.71	9.02
Gender	Man	20	13.2		
	Woman	131	86.8		
Family Status	Married or Cohabitant	106	70.2		
	Single	37	24.5		
	Separated	8	5.3		
Service	ICU	62	41.1		
	Hospitalization	49	32.5		
	Emergencies	26	17.2		
	PC ^1^ or Consultations	14	9.3		
Employment Status	Permanent	77	51.0		
	Interim	43	28.5		
	Temporary Contract	31	20.5		

^1^ PC = primary care.

**Table 2 healthcare-13-01108-t002:** Descriptive analysis of symptoms, burnout syndrome and personality variables (n = 151).

	Mean	SD	95% CI	Median	IQR	Sample Range	Asymmetry	Kurtosis
Anxiety	11.27	5.89	10.32–12.22	11	9	0–21	−0.004	−0.940
Anxiety-2	9.12	5.49	8.24–10.00	8	8	0–21	0.463	−0.509
Anxiety-3	7.80	4.90	7.01–8.59	7	6	0–21	0.548	0.024
Insomnia	12.64	5.95	11.68–13.59	13	9	0–25	−0.036	−0.789
Insomnia-2	10.90	6.17	9.91–11.89	11	9	0–25	0.199	−0.458
Insomnia-3	10.14	6.25	9.13–11.15	10	11	0–25	0.158	−0.862
Emoti. Exhaus. ^1^-2	28.19	13.40	26.04–30.35	29	23	0–54	0.027	−0.858
Depersona. ^2^-2	7.57	6.78	6.49–8.67	6	8	0–27	0.951	0.042
Personal Accom. ^3^-2	35.25	7.80	33.99–36.51	36	11	7–48	−0.842	0.985
Emoti. Exhaus. ^1^-3	23.52	13.79	21.03–25.47	22	17	0–54	0.448	−0.431
Depersona. ^2^-3	9.65	7.03	8.53–10.79	9	12	0–30	0.480	−0.582
Personal Accoml. ^3^-3	29.66	9.97	28.05–31.26	32	13	0–48	−0.616	−0.059
Self-Efficacy	28.96	3.74	28.36–29.56	30	3	18–40	0.028	1.33
Resilience	77.17	14.71	74.81–79.54	81	14	14–98	−1.407	2.347
Social Support	5.80	1.18	5.61–5.99	6	6	1–7	−1.475	2.231
Fear of COVID-2	17.47	6.15	16.48–18.46	17	9	7–35	0.420	−0.066
Cognitive Fusion-2	22.36	10.83	20.62–24.11	22	17	7–49	0.353	−0.860

(2) The number 2 after a variable indicates that the variable was assessed at the second time point. (3) The number 3 after a variable indicates that the variable was assessed at the third time point. ^1^ Emotional exhaustion. ^2^ Depersonalization. ^3^ Personal accomplishment.

**Table 3 healthcare-13-01108-t003:** Analysis of the association between “request for psychological help” and the sociodemographic and occupational variables.

	Mean (SD)	f (%)	Requested Psychological Help	Did Not Request Psychological Help	Statistical Analysis		
					*p* Value	χ^2^	t
Age	41.36 (9.44)		39.35 (9.06)	41.88 (9.50)	0.185		1.33
Years of Experience	10.71 (9.02)		9.61 (8.59)	10.99 (9.14)	0.45		0.758
**Gender**							
Male		20(13.2)	4 (20%)	16 (80%)	0.950	0.004	
Female		131(86.8)	27(21%)	104 (79%)			
**Family Status**							
Married or Cohabiting		106(70.2)	21 (19.8%)	85 (80.2%)	0.924	0.159	
Single		37(24.5)	8 (21.6%)	29 (78.4%)			
Separated		8(5.3)	2 (25%)	6 (75%)			
**Employment Status**							
Permanent		77(51.0)	10 (13.0%)	67 (87%)	0.058	5.695	
Interim		43(28.5)	13 (30.2%)	30 (69.8%)			
Temporary		31(20.5)	8 (25.8%)	23 (74.2%)			
**Service**							
ICU		62(41.1)	9 (14.5%)	53 (85.5%)	0.442	2.691	
Hospitalization		49(32.5)	11 (22.4%)	38 (77.6%)			
Emergency		26(17.2)	7 (26.9)	19 (73.1%)			
PC ^1^ and Consultations		14(9.3)	4 (28.6%)	10 (71.4%)			

^1^ PC = primary care.

**Table 4 healthcare-13-01108-t004:** Association between the request for psychological support and symptoms, burnout syndrome and personality variables.

	Mean (SD)	Requested Psychological Help	Did Not Request Psychological Help	Statistical Analysis	
				*p* Value	Student’s t
Anxiety	11.27 (5.89)	13.30 (4.69)	10.28 (6.19)	0.003	−3.057
Anxiety-2	9.11 (5.49)	11.24 (5.40)	8.07 (5.24)	0.001	−3.428
Anxiety-3	7.80 (4.90)	10.48 (5.13)	6.48 (4.22)	0.000	−4.777
Insomnia	12.64 (5.95)	14.88 (5.85)	11.53 (5.71)	0.001	−3.344
Insomnia-2	10.90 (6.17)	12.76 (6.22)	9.98 (5.96)	0.010	−2.619
Insomnia-3	10.14 (6.25)	12.94 (5.71)	8.75 (6.70)	0.000	−4.153
Self-Efficacy	28.96 (3.74)	27.52 (3.64)	29.67 (3.60)	0.001	3.432
Resilience	77.1 (14.71)	72.60 (15.19)	79.44 (13.99)	0.009	2.67
Social Support	5.80 (1.18)	5.64 (1.11)	5.88 (1.21)	0.222	1.228
Fear of COVID	17.47 (6.15)	17.82 (6.33)	17.30 (6.09)	0.625	−0.490
Cognitive Fusion	22.36 (10.83)	26.58 (10.77)	20.28 (10.29)	0.001	−3.434
Emotional Exhaustion-2	28.19 (13.40)	33.42 (12.40)	25.60 (13.18)	0.001	−3.569
Depersonalization-2	7.58 (6.78)	9.30 (7.59)	6.72 (6.20)	0.040	−2.082
Personal Accomplishment-2	35.25 (7.80)	33.46 (7.17)	36.14 (7.99)	0.040	2.080
Emotional Exhaustion-3	23.52 (13.79)	27.74 (14.10)	21.03 (13.14)	0.006	−2.814
Depersonalization-3	9.65 (7.03)	11.24 (7.82)	8.87 (6.50)	0.068	−1.847
Personal Accomplishment-3	29.66 (9.97)	28.16 (9.49)	30.40 (10.17)	0.186	1.330

(2) The number 2 after a variable indicates that the variable was assessed at the second time point. (3) The number 3 after a variable indicates that the variable was assessed at the third time point.

## Data Availability

Research data will be available upon request to the corresponding author.

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
