# Peer review of "Psychological Care in Spanish Nurses at the Frontline of the COVID-19 Pandemic: A Prospective Study on Symptoms, Burnout and Psychological Variables"

_healthcare, 2025, doi:10.3390/healthcare13101108_

Round 1

Reviewer 1 Report

Comments and Suggestions for Authors

Dear Authors,

I am grateful for the opportunity to review this very interesting manuscript. For quality care in clinical care, it is essential for nurses to have an emotional well-being that helps them to provide optimal care. As is well known, the COVID-19 pandemic has had a devastating impact on the healthcare workforce. This article sheds light on this by clarifying the persistence of psychosocial symptoms resulting from the pandemic and the low demand for help from nurses.  These results are an aid to implementing strategies and interventions aimed at improving nurses' psychological well-being and health promotion, not only as a consequence of the pandemic but also extrapolating to other highly stressful situations or work environments.

- There is a scientific basis in the introduction and the background provided, with extensive bibliography and examples that justify the study.

- In the methodology, when talking about the exclusion criteria, I found it curious to exclude nurses who change service during the study period. Bearing in mind that during this period a large number of professionals were hired and many of these contracts rotated according to the needs of the service, these nurses could surely experience high levels of anxiety, insomnia, Burnout, etc. derived from the situation they were in. What was the reason for not including them?

- The measurement instruments used for data collection add up to a rather large number of items to be addressed. The literature suggests that a questionnaire that is too long can lead to a high non-response rate, could the observed loss in the sample at the three time points be due to this phenomenon? If so, it would be interesting to include this as a limitation of the study.

- The results are correctly presented by reporting the population sample under study, the descriptive characteristics of the participants and the outcome data related to the request for psychological support. In section 3.3 it is reported that only 7.3% (n=11) nurses continued to request psychological support even though they had requested it from the beginning. Why do you think this may be?

- As in the previous section, the discussion analyses and links each of the results to the objectives of the study.

Best regards

Author Response

Psychological care in Spanish nurses at the frontline of the COVID-19 pandemic. A prospective study on symptoms, burnout and psychological variables.

Manuscript ID healthcare-3570693

We would like to thank you for your interest in our manuscript entitled “Psychological care in Spanish nurses at the frontline of the COVID-19 pandemic. A prospective study on symptoms, burnout and psychological variables.” (Manuscript Number: ID healthcare-3570693). We appreciate the time that you and the other reviewers have dedicated to reading the manuscript and providing suggestions. We have incorporated all the comments suggested. Following your suggestions, we have proceeded to revise our manuscript, highlighting the changes with the track changes mode in MS Word. 

At the end of this letter, you will find an explanation of the changes made to the manuscript in accordance with your comments.

We hope the new changes meet their expectations, and we hope that they consider the work apt for publication in Healthcare.

Best regards,

Fernanda Gil

Reviewer 1

- Comment 1: There is a scientific basis in the introduction and the background provided, with extensive bibliography and examples that justify the study.

Response: Thank you very much for your comment. We are delighted to hear that the introduction meets your expectations.

- Comment 2: In the methodology, when talking about the exclusion criteria, I found it curious to exclude nurses who change service during the study period. Bearing in mind that during this period a large number of professionals were hired and many of these contracts rotated according to the needs of the service, these nurses could surely experience high levels of anxiety, insomnia, Burnout, etc. derived from the situation they were in. What was the reason for not including them?

Response: Thank you very much for your comment. We completely agree that it would have been interesting to assess how the change in service could influence increased stress and burnout for nurses. However, within our study, we were interested in assessing which services were most affected by the COVID-19 pandemic and how nursing staff had evolved psycho-emotionally within these services. That is why, in this study, we decided to analyse only staff who had remained in the same service during the data collection period. However, we are very grateful for the suggestion and will consider it as a future line of research, as we do indeed believe that changes in service due to different nursing contracts can be a very significant stress factor.

- The measurement instruments used for data collection add up to a rather large number of items to be addressed. The literature suggests that a questionnaire that is too long can lead to a high non-response rate, could the observed loss in the sample at the three time points be due to this phenomenon? If so, it would be interesting to include this as a limitation of the study.

Response: Thank you very much for the suggestion. We have added the reviewer's suggestion to the limitations section, as we completely agree that a questionnaire with several instruments can take a long time to complete and thus result in a significant loss of sample size.

- The results are correctly presented by reporting the population sample under study, the descriptive characteristics of the participants and the outcome data related to the request for psychological support. In section 3.3 it is reported that only 7.3% (n=11) nurses continued to request psychological support even though they had requested it from the beginning. Why do you think this may be?

Response: We appreciate your comment. We can hypothesise that the decrease in requests for psychological support may be due to different reasons, for example, the stigma associated with requesting psychological support within Latin culture. We can also hypothesise that throughout the data analysis, the lockdown situation in Spain and the fear of COVID-19 improved considerably, allowing people to find support in social and family environments that were not possible at the beginning of the pandemic due to the lockdown situation or fear of the disease itself.

We consider the reviewer's suggestion to be very appropriate and believe it should be incorporated into the discussion in our paper. We have therefore added a paragraph to our paper reflecting on how the percentage of nurses who requested psychological support decreased over the course of the data collection period.

Once again, we thank the reviewer for their suggestion.

- As in the previous section, the discussion analyses and links each of the results to the objectives of the study.

Response: Thank you very much for your comment.

Reviewer 2 Report

Comments and Suggestions for Authors

The article reveals that the demand for psychological support among nurses during the COVID-19 pandemic is high and that this demand is strongly associated with anxiety, insomnia, and burnout. The research emphasizes that it is essential to take measures for crisis management and to strengthen self-efficacy and resilience to improve healthcare professionals' psycho-emotional health. Some deficiencies need to be addressed before this article can be published.

  1. Some numerical or statistical data should be shared in the study's abstract.
  2. Numbers written after keywords should be deleted. A slash should be placed between the word and the number if it is a numerical rating.
  3. Although the study's methodology explains the variables in detail, no detailed information about the method used is provided.
  4. The tables showing the study's statistical results can be explained in more detail.

The study is generally well-designed. This study can be published by addressing the above deficiencies.

Author Response

Psychological care in Spanish nurses at the frontline of the COVID-19 pandemic. A prospective study on symptoms, burnout and psychological variables.

Manuscript ID healthcare-3570693

We would like to thank you for your interest in our manuscript entitled “Psychological care in Spanish nurses at the frontline of the COVID-19 pandemic. A prospective study on symptoms, burnout and psychological variables.” (Manuscript Number: ID healthcare-3570693). We appreciate the time that you and the other reviewers have dedicated to reading the manuscript and providing suggestions. We have incorporated all the comments suggested. Following your suggestions, we have proceeded to revise our manuscript, highlighting the changes with the track changes mode in MS Word. 

At the end of this letter, you will find an explanation of the changes made to the manuscript in accordance with your comments.

We hope the new changes meet their expectations, and we hope that they consider the work apt for publication in Healthcare.

Best regards,

Fernanda Gil.

 Reviewer 2

The article reveals that the demand for psychological support among nurses during the COVID-19 pandemic is high and that this demand is strongly associated with anxiety, insomnia, and burnout. The research emphasizes that it is essential to take measures for crisis management and to strengthen self-efficacy and resilience to improve healthcare professionals' psycho-emotional health. Some deficiencies need to be addressed before this article can be published.

Comment 1: Some numerical or statistical data should be shared in the study's abstract.

Response: We sincerely thank the reviewer for this valuable suggestion. In response, we have revised the abstract to include specific numerical data that emphasize the statistical relevance of our findings. In particular, we have incorporated percentages illustrating the proportion of nurses who requested psychological support during the study, as well as the significant associations observed between the request for psychological help and key emotional symptomatology (such as anxiety and insomnia) and personality-related variables (such as resilience and self-efficacy). These additions strengthen the abstract by providing a more concrete and informative summary of our results.

Comment 2:  Numbers written after keywords should be deleted. A slash should be placed between the word and the number if it is a numerical rating.

Response: We thank the reviewer for this helpful suggestion. In accordance with the recommendation, we have removed the numbers that appeared after the keywords. 

Comment 3: Although the study's methodology explains the variables in detail, no detailed information about the method used is provided.

Response: We thank the reviewer for this comment. In response, we have revised the "Materials and Methods" section to explicitly clarify the methodological approach of our study. Specifically, we now indicate that the study followed a quantitative, prospective longitudinal observational design, with data collection based on validated self-report instruments. Furthermore, we noted that the study was conducted in accordance with STROBE guidelines to enhance transparency and rigor.

Comment 4: The tables showing the study's statistical results can be explained in more detail.

Response: We sincerely thank the reviewer for this helpful suggestion. In response, we have expanded the description of the statistical results presented in Tables 3 and 4. Specifically, we have provided more detailed narrative explanations within the results section, outlining the distributions, group comparisons, and statistical significance of the variables in relation to the request for psychological support. We believe these additions enhance the clarity and depth of the findings.

Reviewer 3 Report

Comments and Suggestions for Authors

Dear Authors,

This manuscript presents a prospective longitudinal observational study conducted over three time points (2020–2022) to examine psychological symptoms, burnout, and personality variables among Spanish nurses working during the COVID-19 pandemic. The topic is highly relevant and timely, addressing the mental health impact of the COVID-19 pandemic on frontline healthcare workers and the longitudinal design provides valuable insight into the progression of psychological symptoms. However, I have some concerns related to Methods section:

- The sample was collected through convenience sampling and is heavily concentrated in the Madrid and Catalonia regions. This significantly limits the national representativeness of the findings and may bias generalizability.

- The sample decreased from 534 participants at baseline to 151 at the final time point (~72% attrition). There is no attrition analysis to compare responders and non-responders, introducing potential survivorship bias.

- Psychological help-seeking behavior was assessed through a single ad-hoc yes/no question, with no details on the type, frequency, duration, or actual receipt of psychological support. This may underrepresent the complexity of mental health service utilization.

- There is no control for pre-existing psychological conditions or relevant confounders (e.g., prior trauma, pre-pandemic mental health), which may affect the observed associations.

- Not all variables were measured at all time points (e.g., burnout only at T2 and T3), which limits longitudinal analysis and comparability. The rationale for this is not explained.

- Organizational and institutional factors (e.g., access to mental health resources, workload, leadership support) are not assessed, despite their well-documented impact on burnout and help-seeking behavior.

These aspects should be considered within the scope of the study's limitations. Additionally, it may be beneficial to introduce multivariate regression models to predict psychological help-seeking behavior, while controlling for relevant covariates. It would also be pertinent to discuss the practical implications of the findings for workplace mental health interventions and policies, particularly those grounded in protective factors such as resilience and self-efficacy. Furthermore, providing a clear justification for the chosen time-point measurements and ensuring consistency across study waves would strengthen the methodological rigor.

Author Response

Psychological care in Spanish nurses at the frontline of the COVID-19 pandemic. A prospective study on symptoms, burnout and psychological variables.

Manuscript ID healthcare-3570693

We would like to thank you for your interest in our manuscript entitled “Psychological care in Spanish nurses at the frontline of the COVID-19 pandemic. A prospective study on symptoms, burnout and psychological variables.” (Manuscript Number: ID healthcare-3570693). We appreciate the time that you and the other reviewers have dedicated to reading the manuscript and providing suggestions. We have incorporated all the comments suggested. Following your suggestions, we have proceeded to revise our manuscript, highlighting the changes with the track changes mode in MS Word. 

At the end of this letter, you will find an explanation of the changes made to the manuscript in accordance with your comments.

We hope the new changes meet their expectations, and we hope that they consider the work apt for publication in Healthcare.

Best regards,

Fernanda Gil

Reviewer 3

Comment 1: The sample was collected through convenience sampling and is heavily concentrated in the Madrid and Catalonia regions. This significantly limits the national representativeness of the findings and may bias generalizability.

Response: We thank the reviewer for this important observation. As correctly noted, the sample was collected using a non-probabilistic convenience sampling method, which did not allow us to control for geographical distribution. Consequently, most responses were concentrated in the Madrid and Catalonia regions, limiting the representativeness of the sample at the national level. This limitation is acknowledged in the manuscript.

This sampling method was chosen due to the exceptional circumstances at the time of the study’s initiation, when Spain was under strict lockdown measures because of the high incidence of COVID-19. These conditions made it extremely difficult to access the healthcare workforce directly. For this reason, the survey was disseminated primarily through social media and institutional emails to reach the greatest number of healthcare professionals. Although the intention was to access a wide national population, responses were predominantly received from Madrid and Catalonia, likely because these were among the regions most affected by the pandemic in its early stages. This suggestion has been fully taken into account and is now included as a limitation in the corresponding section of the manuscript.

Comment 2: The sample decreased from 534 participants at baseline to 151 at the final time point (~72% attrition). There is no attrition analysis to compare responders and non-responders, introducing potential survivorship bias.

Response: We thank the reviewer for this important observation. We acknowledge that the study experienced a high attrition rate, with a reduction from 534 participants at baseline to 151 at the final time point. This substantial dropout is a known limitation of longitudinal studies involving healthcare workers, particularly under the challenging conditions of the COVID-19 pandemic. During the study period, participants faced high workloads, emotional exhaustion, and limited time availability, all of which may have contributed to reduced participation in follow-up assessments.

We recognize that this limitation may introduce survivorship bias, and we have explicitly acknowledged this in the limitations section of the manuscript. We appreciate the reviewer’s suggestion, which has helped us improve the transparency of our study's methodological limitations.

However, in relation to the responders/non-responders analysis, it should be noted that no statistically significant differences were observed in the variables of interest between the nurses who did not complete the three phases of the study (n=383) and those who completed all the phases of the study (n=151). This information has been included in the revised version of the manuscript.

Comment 3: Psychological help-seeking behavior was assessed through a single ad-hoc yes/no question, with no details on the type, frequency, duration, or actual receipt of psychological support. This may underrepresent the complexity of mental health service utilization.

Response: We sincerely thank the reviewer for this thoughtful observation. Indeed, psychological help-seeking behavior in our study was assessed through a single ad-hoc yes/no question, and we did not gather information regarding the type, frequency, duration, or actual receipt of psychological support. We acknowledge that this approach may not fully capture the complexity and nuances of mental health service utilization. Given the relevant associations observed between help-seeking and both symptomatology and personality variables in our results, we have identified this limitation as a valuable area for future research. We intend to explore these aspects in greater detail in future studies to better understand patterns and effectiveness of psychological support among healthcare professionals.

Comment 4: There is no control for pre-existing psychological conditions or relevant confounders (e.g., prior trauma, pre-pandemic mental health), which may affect the observed associations.

Response: We thank the reviewer for this valuable observation. We acknowledge that the study did not control for pre-existing psychological conditions or other relevant confounders such as prior trauma or pre-pandemic mental health status. We agree that these factors could influence the associations observed in our results. This limitation has been explicitly acknowledged and discussed in the limitations section of the manuscript, as we consider it a potential source of bias in the study.

Comment 5: Not all variables were measured at all time points (e.g., burnout only at T2 and T3), which limits longitudinal analysis and comparability. The rationale for this is not explained.

Response: We thank the reviewer for this insightful comment. Indeed, not all variables were assessed at every time point. Specifically, variables such as anxiety and insomnia were measured across all three phases in order to capture their peak and evolution over time. In contrast, burnout was conceptualized by the research team as a potential consequence of prolonged exposure to pandemic-related stressors and, therefore, was assessed starting from the second time point. We agree that having baseline data on burnout would have enriched the analysis and allowed for a more rigorous evaluation of its progression throughout the study period. This limitation has now been acknowledged and incorporated into the limitations section of the manuscript.

Comment 6: Organizational and institutional factors (e.g., access to mental health resources, workload, leadership support) are not assessed, despite their well-documented impact on burnout and help-seeking behavior.

Response: We thank the reviewer for this important and well-founded observation. We acknowledge that the study did not include organizational and institutional variables such as access to mental health services, workload, or leadership support, which are known to influence both burnout levels and help-seeking behavior among healthcare professionals. We fully agree that the absence of these contextual factors is a relevant limitation, and we have now incorporated this point into the limitations section of the manuscript.

Comment 7: These aspects should be considered within the scope of the study's limitations. Additionally, it may be beneficial to introduce multivariate regression models to predict psychological help-seeking behavior, while controlling for relevant covariates. It would also be pertinent to discuss the practical implications of the findings for workplace mental health interventions and policies, particularly those grounded in protective factors such as resilience and self-efficacy. Furthermore, providing a clear justification for the chosen time-point measurements and ensuring consistency across study waves would strengthen the methodological rigor.

We sincerely thank the editor for this insightful and constructive comment. We have carefully considered each of the points raised and incorporated them into the revised manuscript as follows:

First, we have acknowledged the study’s limitations in greater depth, including the lack of assessment of pre-existing psychological conditions, the use of a convenience sample with geographic concentration, and the absence of organizational-level variables. These limitations are now clearly addressed in the discussion section.

In response to the editor's suggestion, in order to analyse, among the possible statistically significant variables at bivariate level, the predictor variables within the overall impact, a logistic regression analysis was carried out (enter method). The results show that anxiety experienced at the third time point is the only predictor variable (B=.225, SD=.047; Wald=23.153, Exp (B)=1.252, p<.001). This new finding has been incorporated into the revised version of the manuscript.

Furthermore, we have expanded the discussion to reflect on the practical implications of our findings, particularly regarding the importance of reinforcing protective psychological traits such as resilience and self-efficacy in institutional interventions and mental health support strategies for healthcare professionals.

Lastly, we have added a justification for the choice of time-point measurements. Specifically, anxiety and insomnia were measured longitudinally to examine their evolution across the pandemic, while burnout was assessed at later stages due to its conceptualization as a cumulative response. We acknowledge the methodological limitations of this approach and have discussed it accordingly.

Once again, we are grateful for this recommendation, which has helped us enhance the analytical and interpretive rigor of the study.

Round 2

Reviewer 3 Report

Comments and Suggestions for Authors

Congratulations on the work developed.